# Synergies and Trade-Offs between Lean-Green Practices from the Perspective of Operations Strategy: A Systematic Literature Review

**Geandra Alves Queiroz** [1,*], **Ivete Delai** [2], **Alceu Gomes Alves Filho** [2], **Luis Antonio de Santa-Eulalia** [3] **and Ana Lúcia Vitale Torkomian** [2]

1   Department of Engineering, Production Engineering, University of Minas Gerais State (UEMG), Passos 37900-106, MG, Brazil
2   Department of Industrial Engineering, Federal University of São Carlos, São Carlos 13565-905, SP, Brazil
3   Department of Information Systems and Quantitative Methods in Management, École de Gestion, Université de Sherbrooke, Sherbrooke, QC J1K 2X9, Canada
*   Correspondence: geandraqueiroz@gmail.com

**Abstract:** In the operations management and sustainability literature, the integration of Lean and Green manufacturing is considered one of the great solutions to balancing operational gains and environmental sustainability. This literature focuses mainly on the integration between them. However, there are no studies investigating how this integration is related to the Operations Strategy content: competitive priorities and decision areas. Thus, this study aims to contribute to reducing this research gap by providing a more in-depth understanding of the relationships between Lean-Green practices from the point of view of the Operations Strategy. We identify synergies and potential trade-offs between competitive priorities and changes in decision areas when Lean-Green practices are implemented. We performed a systematic literature review to answer two questions: Does the implementation of Lean and Green practices affect operations' competitive priorities, causing synergies or trade-offs? What decision area(s) are modified with the implementation of each practice? This systematic review analyzed 338 selected articles. Competitive priorities, decision areas, Lean practices, Green practices and Lean-Green practices were identified and discussed, highlighting trade-offs, synergies and changes in decision areas. The results suggest that Lean and Green are synergistic in most practices, but they must be managed according to the Operations Strategy, especially as their focuses are essentially different and trade-offs may occur.

**Keywords:** lean manufacturing; green manufacturing; competitive priorities; decision areas; sustainability

## 1. Introduction

Environmental problems, such as climate change, pollution, the reduction of natural resources and loss of biodiversity, tend to collapse the planet [1]. Facing this context, society, governments, investors, and companies themselves have increasingly demanded the elimination or reduction of the environmental impacts of products and production processes [2,3]. To meet these demands, companies seek to adopt programs and practices that reduce such impacts but, at the same time, provide for the achievement of their competitive production priorities [4,5]. An Operations Strategy (OS) aligned with market requirements is essential, as it can determine the company's competitive advantage [6].

Lean and Green Manufacturing practices have been seen as a solution to improve and balance all the competitive priorities of OS; the relationship between these two strands has been explored by the Lean-Green (LG) integration literature. The study of Leong et al. [6] pointed out that the LG approach can obtain the maximum operational performance without compromising the environment. Recent research [7,8] presents cases from results in

both operational gains and environmental aspects, such as lead time reduction, and water and wastes reduction. Complementary, Caldera et al. [9] consider that LG enable the transition to sustainable business. Many studies [10–12] demonstrate that Lean aims to reduce wastes in the value chain, which can contribute to reducing costs and defects and to the increasing of natural resources (e.g., water, energy and materials) efficiency. Lean and Green are complementary, and Lean enables the development of environmental management capability helping to "green" the organization [13–15]. However, Lean does not take into account environmental impacts directly. Thus, organizations need to implement Green tools into the Lean management to fill this gap [14]. To refer to Lean and Green integration, the term "Lean-Green" has been used [16–19] and will be adopted in this work.

Research about the Lean-Green has addressed three main topics. Some studies explore the relationship between Lean and Green; they present the synergies and differences among them. The authors argue that the approaches are very compatible but some trade-offs may appear and should be considered [13,20–29]. Another group of authors proposes tools and frameworks to integrate Lean and Green, describing requirements and steps to implement them, as well as barriers and enablers to doing so [2,14,18,22,30–40]. There is also a group focused on the implementation of LG. In general, these studies show improvements in cost reduction, quality and environmental performance; mainly through reduction in energy consumption and waste generation [7,10,41–43]. However, there are studies that show some negative results in the environment, such as the increase of emissions when Just-in-Time (JIT) is implemented [13,20,44–50].

Despite these efforts, an integrated and holistic understanding of how LG are linked with all OS content (namely, competitive priorities and decision areas) is still missing. Chatha and Butt [51] presented an extensive literature review about OS, providing a historical overview and the current status of this topic; but the results did not show any study that discussed LG and the entire OS content. There are only a few studies in the literature that somehow correlated OS with LG: Longoni and Cagliano [52] provided evidence about how the cross-functional executive involvement and worker involvement, in the formulation and implementation of the OS supporting the strategic alignment of Lean and sustainability; Suifan et al. [53] analyze the trade-offs between LG and through a multi-criteria decision-making shows that competitive priorities can differ in each approach; and, Queiroz et al. [26] present the competitive priorities and the Lean-Green practices adopted for automotive suppliers However, both studies still do not provide a wide understanding about the relationships between LG and OS.

In addition, it is important to mention there are some systematic reviews that focus on Lean-Green [4,36,54–59]. However, these studies do not consider the relationship of Lean-Green and Operations Strategy content (competitive priorities and decision areas) to understand the trade-offs, synergies and changes in the decision areas. Considering the relevance of the operations function–for the organization's competitiveness, the current stakeholders' requirements, and the lack of studies on LG practices from the perspective of the consolidated background of OS–this study aims to provide a broad perspective on the possible impacts of adopting LG practices on the content of OS, highlighting the occurrence of synergies and tradeoffs between competitive priorities and the changes promoted in decision areas. Thus, this work seeks to take a step towards systematizing contributions from studies that have addressed issues related to constructs in the field of OS—competitive priorities and decision areas—when LG practices are implemented, and to update and expand this part of part of a doctoral thesis developed in 2021 by Queiroz et al. [60]. With this objective, a Systematic Literature Review (SLR) and a content analysis of 338 articles were carried out from 1996 to the present moment. The results can contribute to a better understanding of the trade-offs and implementation practices aligned with corporate sustainability objectives, allowing the development of well-positioned production systems that can meet new market demands and be consistent with the global strategy of the company.

This paper is organized as follows. We present a brief theoretical background about OS and LG in Section 2. Next, Section 3 explains the research design, the process of data collection and the method used to select studies and perform content analysis. Subsequently, Section 4 discusses our findings, which include descriptive evidence regarding the sample of articles and the results from the content analysis to answer the research questions. Lastly, Section 5 highlights the main implications, and proposes some avenues for future research.

## 2. Theoretical Background

Operation Strategy is the set of decisions that seeks to balance production resources with market needs to contribute to the overall strategy of organizations [61]. Skinner [62] published the first study discussing OS; it was emphasized that the production function should be considered strategic and as a source of competitive advantage. The implementation of an adequate OS, including the development of production function capabilities, plays a crucial role for companies in the business environment, and must be in line with the way the company seeks to create competitive advantages [63].

The success of an OS is related to the definition of its content, which is composed of competitive priorities (CP) and actions to be implemented in decision areas (DA) [64]. CP are related to the performance objectives that the production function adopts to align itself to the company's competitive strategy [62], which are: cost, quality, delivery, flexibility and service [62,65]. Moreover, as indicated by Longoni and Cagliano [5], "environment" can be considered another competitive priority of the operations. These priorities are achieved through a pattern of decisions and actions implemented in the set of DA of the company, such as facilities, capacity, technology, supply chain, human resources, quality, production planning and control, product development, performance measurement systems and organization [62,66].

The content of OS can be seen through the lens of the "strategic choices" paradigm (one of the three proposed by [64,67] in which strategic decisions in the processes and infrastructures of organizations guide the implementation of practices (or actions) and changes in decision areas aimed at improving the performance of operations and gaining competitive advantages. This is the perspective chosen in this work to examine the impacts promoted by the adoption of LG practices in decision areas and competitive production priorities.

Lean Manufacturing (LM) is considered one of the most used approaches to improve operations performance and increase competitiveness [68]. LM is a set of principles and practices that aims to eliminate all kinds of waste in an organization [69]. It is an approach that goes beyond a production management strategy, and it can be considered a management philosophy [70] and an integrated socio-technical system [71]. The main LM practices are 5s, Kaizen, Value Stream Map (VSM), Just in Time (JIT), SMED, Total Productive Maintenance (TPM), Kanban, Standardized Work, Visual Management and 5 Why's (root cause analysis) [72].

Regarding the concept of Green Manufacturing (GM) or Sustainable Manufacturing (SM), it emerged in the 1990s as a philosophy and operational approach to reduce the negative environmental impacts of products [4]. It concerns the search to reduce pollution, energy consumption and the generation of toxic substances through the development of new processes in the manufacturing phase [73–75]. According to research [75,76], GM, or SM, encompasses different tools to reduce the environmental impacts generated by production processes, such as: Cleaner Production, (Life Cycle Assessment (LCA), Environmental Management System (EMS), Circular Economy (CE), Eco-design/Design for Environment, Green/Sustainable Supply Chain, and 3R (Recycling, Remanufacturing and Reuse) [75].

LG integration has been considered the approach that supports achieving the sustainability performance (economic, environmental and social dimensions) of a production system [77]. There are many proposals in the literature, like frameworks for LG integration, cases that show positive and negative environmental results from Lean implementation, and some integrated tools such as Environmental Value Stream Mapping—E-VSM [78], 7s—that is, 5s plus S (safety) and S (sustainability) [79], and Green Lean Six Sigma [33]. Figure 1 presents the constructs that will be discussed in this SLR.

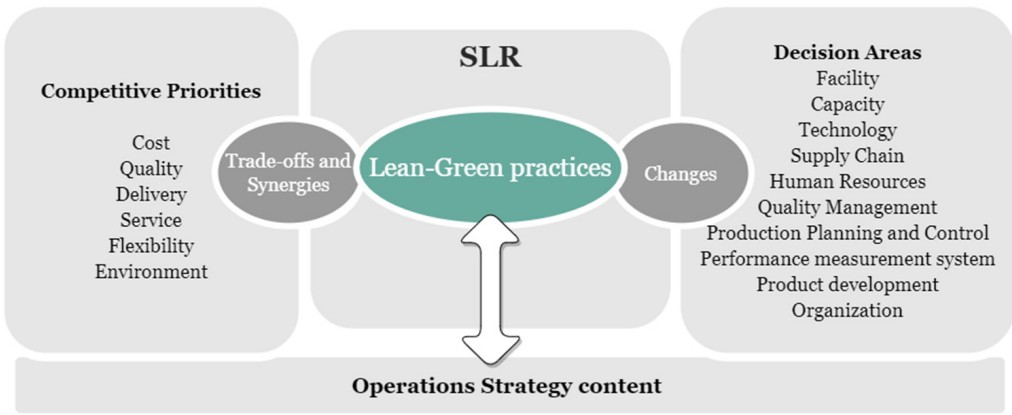

**Figure 1.** Relationships between the concepts, research method and results. Source: created by authors.

A SLR was carried out to identify contributions in the literature that highlighted the impacts of the implementation of LG practices on operations' competitive priorities and on decision areas.

## 3. Research Design

This SLR followed the three macro stages proposed by Denyer and Tranfield [80], as well as the Prisma Statement Flow Diagram proposed by Moher et al. [81]. The SLR process is detailed in Figure 2, which illustrates the summary of the SLR protocol to ensure the transparency and reliability of the process. Initially, the SLR protocol was elaborated and validated jointly by all authors. Throughout the SLR development, meetings between the authors were held to evaluate the results and resolve any disagreement.

### 3.1. Research Question Formulation

We established the SLR research question needed to achieve the aim of the project, which was to understand how Lean-Green are related to the OS content. Considering this, the research questions addressed in this review are:

**RQ1.** Does the implementation of Lean and Green practices affect operations' competitive priorities, causing synergies or trade-offs?

**RQ2.** What decision area(s) are modified with the implementation of each practice?

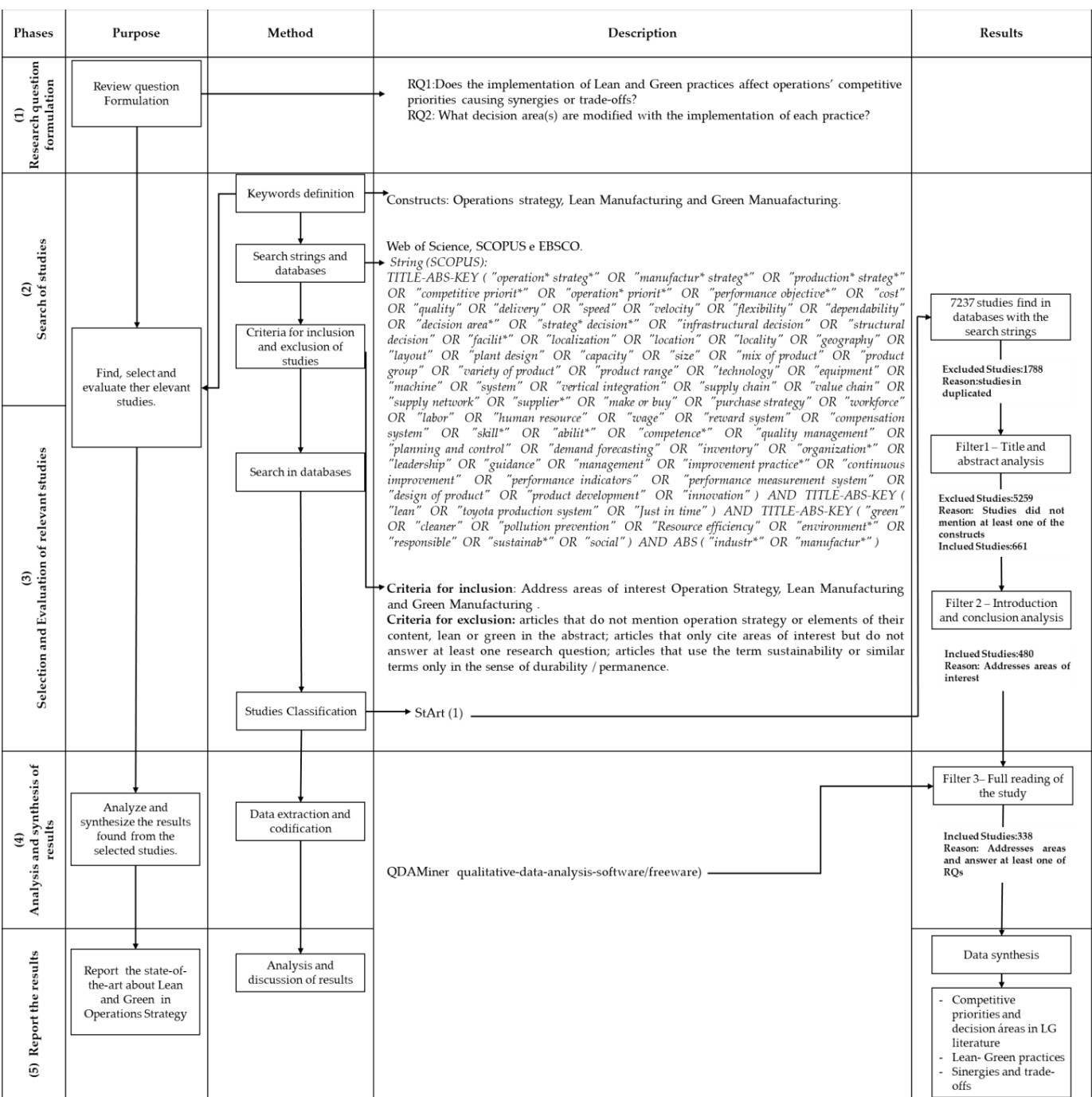

**Figure 2.** Summary of SLR Protocol. Source: created by authors. http://lapes.dc.ufscar.br/tools/start_tool (accessed on 21 November 2018). https://provalisresearch.com/products/qualitative-data-analysis-software/ (accessed on 10 July 2019).

*3.2. Search Strategy*

Studies were searched in three databases chosen according to their scientific scope to provide better results. The Scopus from Elsevier and The Web of Science from Thomson Reuters Institute of Scientific Information were chosen as they are regularly updated and have a wide breadth of coverage in most scientific subjects [82]. Seeking to improve the scope of searches, we also included the EBSCO because it is an extensive database in management.

After a preliminary review of the literature in OS [5,51,52,62,83], LM [3,72] and GM [73,74,84], the strings were developed to conduct the search in the databases. We covered several keywords related to the constructs from the RQs: OS, CP, DA, LM and GM. The search method consisted of the use of strings, defined in such a way as to return results that simultaneously contained at least one keyword referring to each construct. In the search string we considered all the synonyms of the constructs, and we detailed all the CPs and the DAs. A more detailed string was chosen for a wider range of articles since there are articles that focus only on one CP or one DA. Furthermore, we did not limit the field "year" to obtain the largest number of articles on the theme. The search in the databases was first carried out in January 2019 and then updated in November 2020 and again in February 2023. In all databases, we did the search in title, abstract and keywords by focusing on journal articles (excluding books and conference papers). Figure 2 presents the strings used in the searches and the summary of the SLR protocol.

### 3.3. Selection and Evaluation of Relevant Studies

Studies were selected through three filters. The initial search resulted in 7237 studies; it is worth mentioning that 1788 duplicates were excluded. In sequence was applied to the first Filter with the reading of titles and, when necessary, the reading of abstracts. This filter was applied to 5449 papers with the support of StArt software version 3.0.3 BETA. The process was manual, and it took two months to be concluded. There were studies that were very simple to exclude because some titles were completely out of the domain area researched; and for titles that were related with RQs, it was necessary to read the abstract and analyse them. To select the articles for this review, we applied two sets of inclusion and exclusion criteria, presented in Figure 2. The first one was applied during the screening phase; it searched title, abstract and keywords, and included articles that presented at least one term of each construct, related with content of the OS, LM and GM. For example, a document that presented one CP, like "quality", one Lean practice (LP), like "Kanban", and one Green practice (GP), like "Life Cycle Analysis", was selected for the next step. The articles that did not mention at least one decision area or competitive priority or any GP, were excluded.

Then, in Filter 2 (eligibility) we read the (each) paper's introduction and conclusion, and included those that fulfilled the search inclusion criteria: full content access, written in English, published in scientific peer-reviewed journals and discuss at least one element of each construct. For example, the study of [85] cites "Lean" but does not discuss any practice, referring only to some aspect of OS and GM. We also found cases of using the term "sustainability" just to refer to the stability of the practices implemented, as the study of ref. [86]. Studies like these were also excluded following the exclusion criteria "Articles that use the term sustainability or similar terms only in the sense of durability/permanence".

The third filter was then applied to the full paper using the same inclusion and exclusion criteria as Filter 2. In this filter, 142 papers were excluded. As a result, a total of 338 were selected for content analysis to answer the proposed RQs. The main reason for the excluded papers was because the papers mentioned the three constructs but did not answer the research questions. One example is the study of ref. [87] that only cites the LP "JIT" as an example of an initiative in operations. Therefore, considering this sample of 338 papers, the next topic will present how the analysis of this work was done.

### 3.4. Analysis and Synthesis of the Results

After Filter 3, an analysis of 338 papers was done in full, aimed at extracting specific information from studies related to the research topic. In this filter, the articles were analyzed in a descriptive manner, seeking to generate a classification of the articles by year, journal, country of empirical studies, research method, industrial sector, and main research focus. Moreover, a content analysis was made by following the recommendations of [88] seeking to answer the research questions (RQs). The QDA Miner Software (Version 5) was used as a tool to facilitate the analysis process (individual papers and cross-papers).

According to ref. [89] et al., this software collaborates in the organization of ideas and comparison between the cited. The analysis steps are presented in Figure 3.

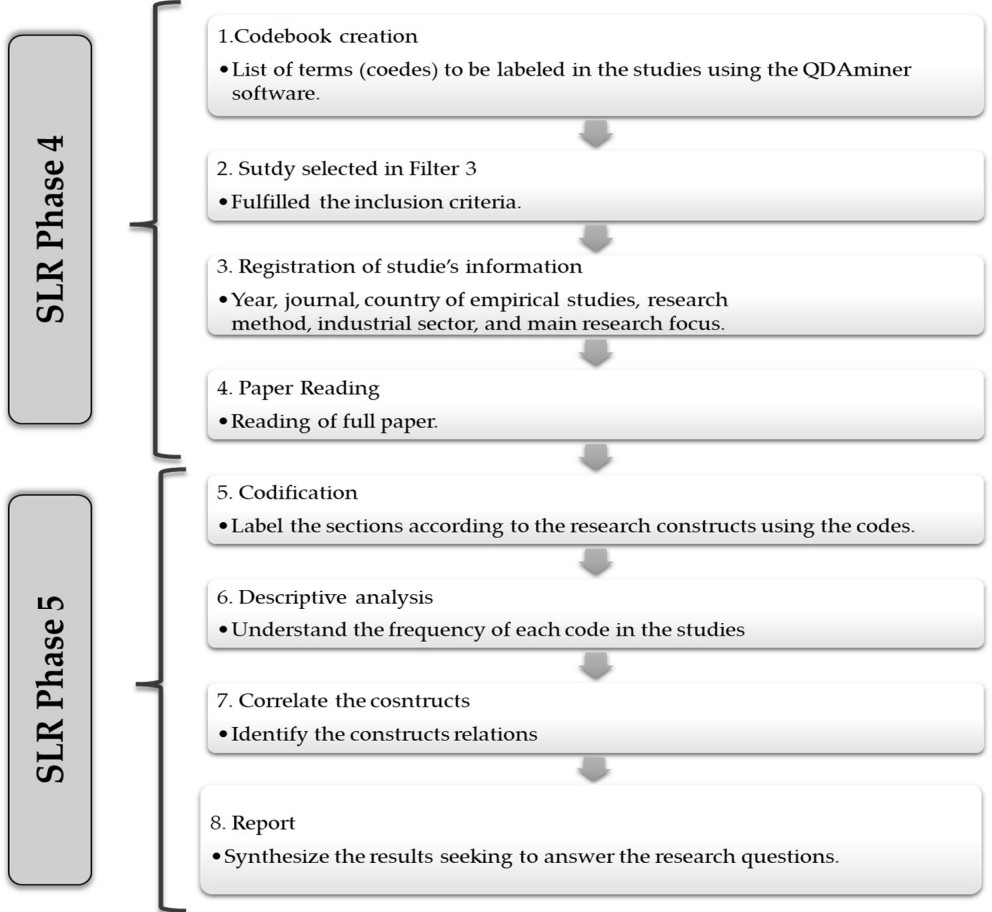

**Figure 3.** SLR Analysis steps.

The data were coded following the basic requirements proposed by Krippendorff [88]. The codes are very important to help identify relationships and establish connections among the many studies that write about the same topics [72,90]. We used the concept-based coding, extracting data from the texts and others that emerged from the reading related with database coding, as suggested by Gibbs [90]. The codebook, attached in the supplementary material, was defined based on the constructs found in the literature of OS content (competitive priorities and decision areas), LM and GM. In this codebook we specified what the initial codes are from the literature and those from the final process.

The content analysis process started looking for the LP and GP, and the CP and DA in the studies. Later, the frequency analysis of the constructs was carried out seeking which LG practices and elements from OS are discussed in the LG literature. This whole process was supported and supervised by three senior researchers. Once the papers were codified, and the content of OS and the practices from LM and GM were identified, it was possible to find the relationship between them all. In the supplementary material are specified all the competitive priorities, decision areas, lean practices, green practices and lean-green practices found and studied where they were addressed. The next topic presents the analysis results (Step 5 of SLR) and discussion.

## 4. Results and Discussion

The results concerning LG from the perspective of OS are discussed in two parts. First, we describe the sample (Section 4.1), and then how Lean-Green is related to the OS content (Section 4.2).

### 4.1. Descriptive Analysis

It is observed that half of the studies were published in the last four years (Figure 4). The first two publications were Florida [12] and Ferrone [44], and the year with the largest number of publications was 2019. This growth may be attributed to two main reasons. First, the need to integrate sustainability issues in productive systems has awakened the interest of academia in studying practices that focus on that. Secondly, based on the initial proposals of LG integration, studies have focused on their validation and implementation.

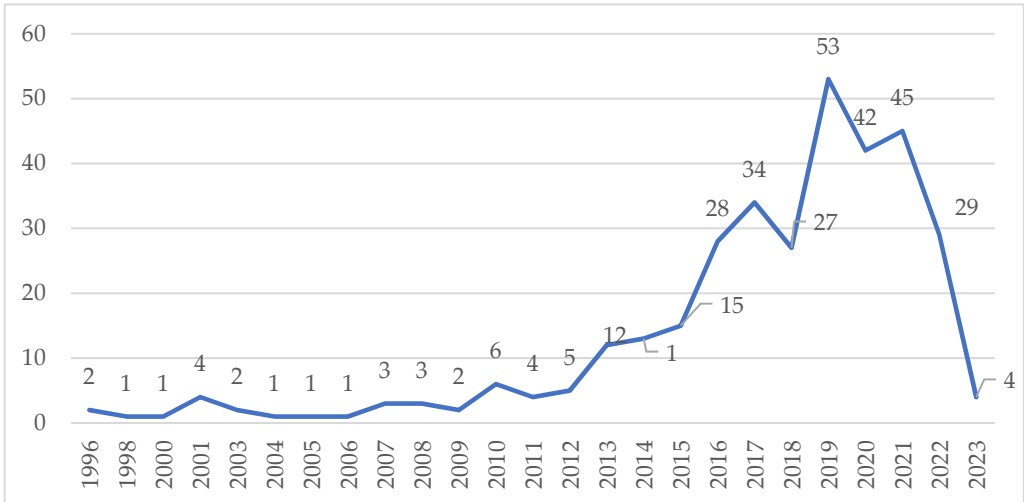

**Figure 4.** Historical evolution of articles analyzed. Source: created by the authors.

In terms of the number of publications per journal, studies were found in 153 different journals, and the journals with more than 1% of publications presented in Figure 5 50.15% of the studies analyzed. *The Journal of Cleaner Production* is the outlet with the highest number of publications on this topic, with 61 publications, which corresponds to 18% of the total articles. The second journal is *Sustainability* with 5.93%. Figure 5 presents this ranking and the journals that have up to three articles.

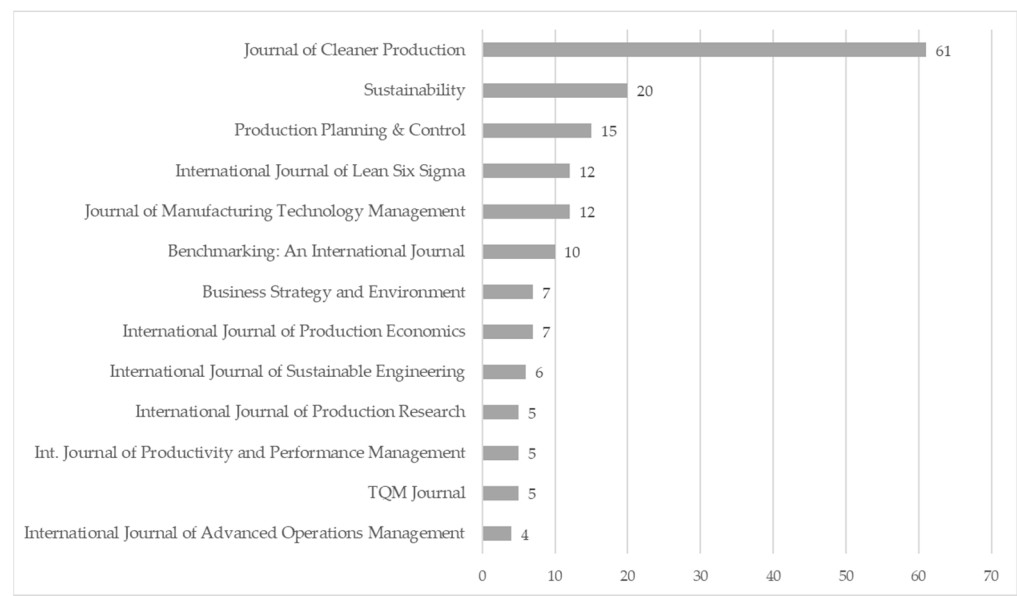

**Figure 5.** Number of articles published by journal. Source: created by the authors.

Figure 6 presents the research methods used in the studies and some characteristics of the samples for the empirical papers. There is a rising trend to adopt empirical studies (79.88%) that applied mainly case studies (33.72% of the sample) and survey (26.03%).

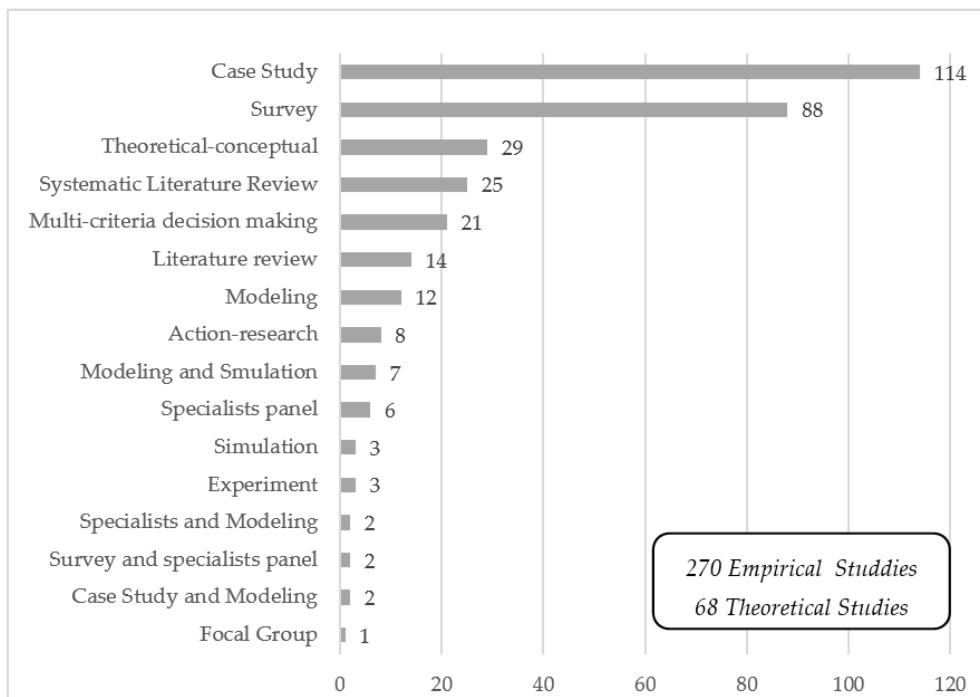

**Figure 6.** Research Method.

Furthermore, Figure 7 presents the classification of the empirical studies regarding the industrial sector and the country, where the study was done and the size of the organizations. Regarding the industrial sector where empirical research was done, the three most frequent sectors were automotive, metal-mechanical, civil construction and electro-electronics, corresponding to 36.66% of the empirical studies.

As for the location, the majority (around 45%) of the research was done in India, Brazil, Malaysia, the United States and China. UNCTAD (United Nations Conference on Trade and Development) [91] developed the status classification. Only the US is classified as a developed economy, while the other ones are classified as developing economies. Therefore, these data can mean that countries that focus on developing their economies are seeking practices to improve competitiveness sustainably. Figure 8 presents a global map with the papers' distributions, and Table 1 presents a distribution of the papers per country of empirical studies.

Regarding the size of the companies in the empirical studies, Figure 9 representd that only 100 studies identified it, with 12% studying large companies, 9.62% studying SMEs companies, and 4% the small companies; 7.4% were done with a mixed sample including small, medium and large, and another 2.2% represented medium companies.

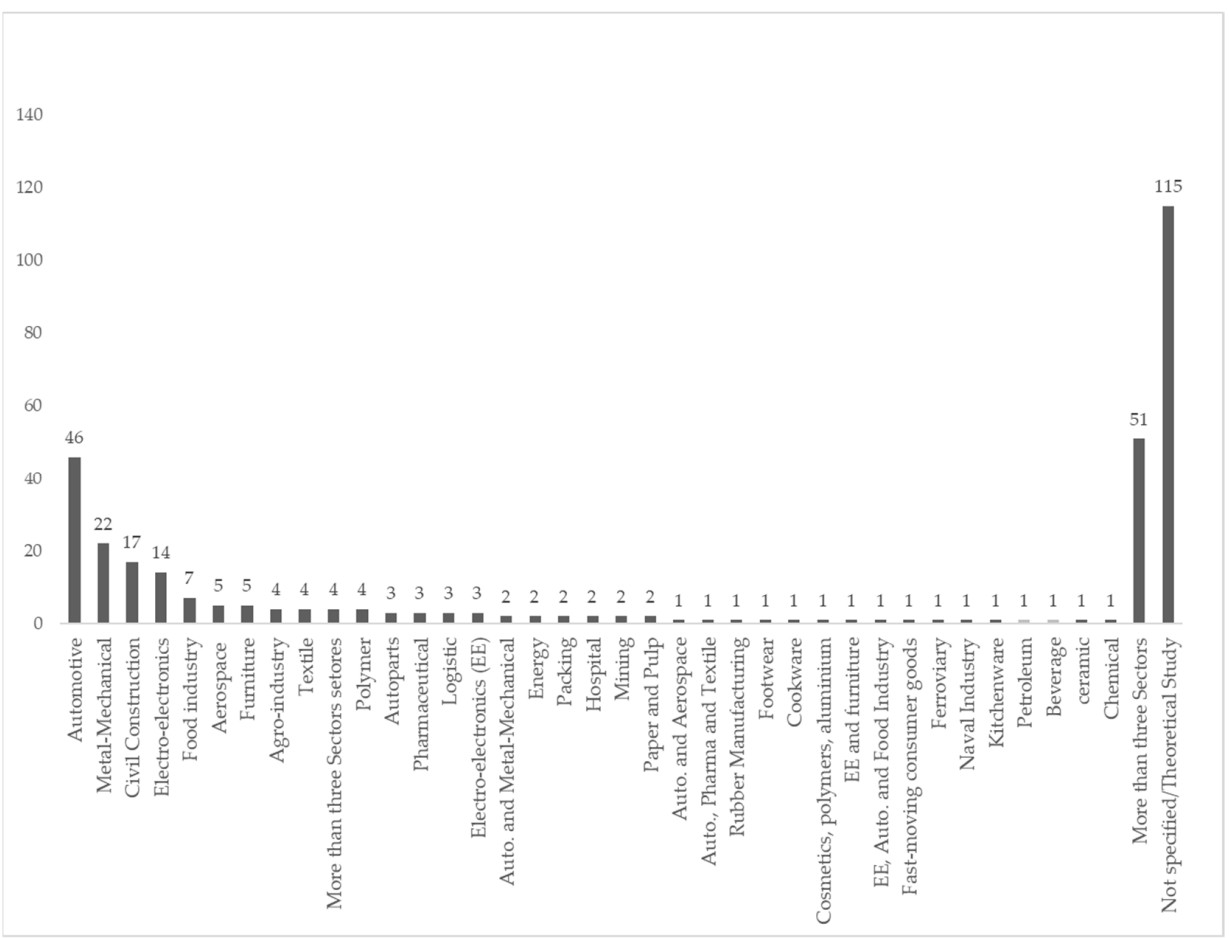

**Figure 7.** Industrial Sector of empirical studies.

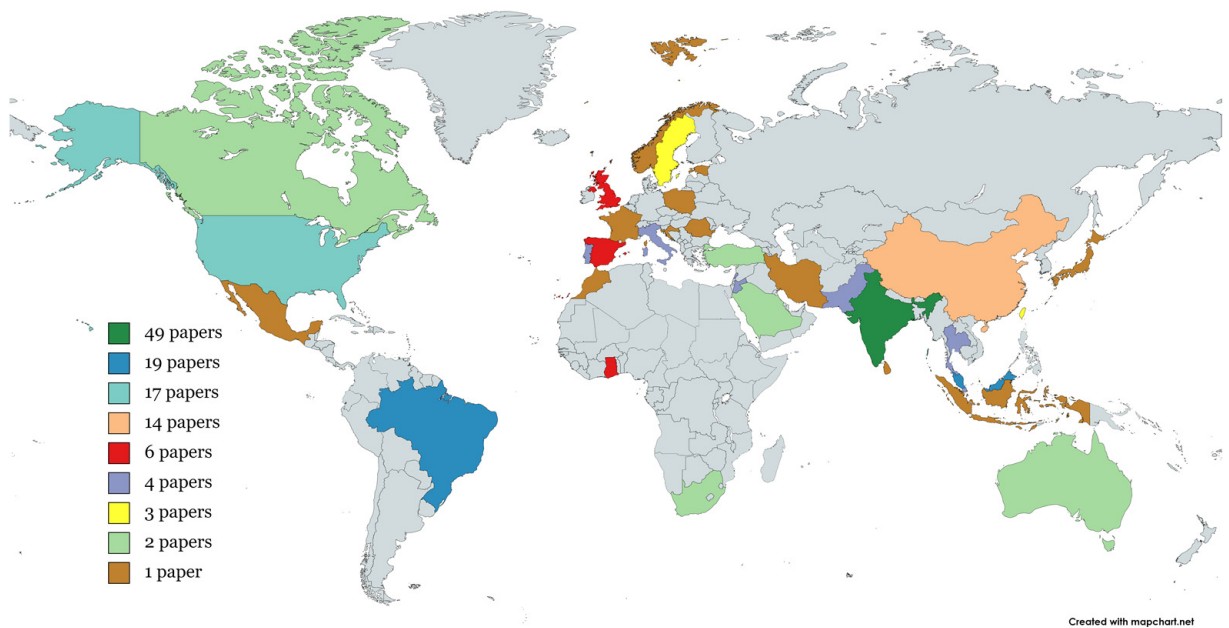

**Figure 8.** Papers per country of empirical studies.

**Table 1.** Distribution of the papers per country of empirical studies.

| Country of Study | Total of Studies | Total of Studies (%) |
| --- | --- | --- |
| Not specified or theoretical study | 136 | 40.24% |
| India | 49 | 14.50% |
| Brazil | 19 | 5.62% |
| Malaysia | 19 | 5.62% |
| United States | 17 | 5.03% |
| China | 14 | 4.14% |
| Several countries | 9 | 2.66% |
| United Kingdom | 6 | 1.78% |
| Spain | 6 | 1.78% |
| Ghana | 6 | 1.78% |
| Portugal | 4 | 1.18% |
| Singapore | 4 | 1.18% |
| Jordan | 4 | 1.18% |
| Pakistan | 4 | 1.18% |
| Thailand | 4 | 1.18% |
| Italy | 4 | 1.18% |
| Sweden | 3 | 0.89% |
| Taiwan | 3 | 0.89% |
| Australia | 2 | 0.59% |
| Canada | 2 | 0.59% |
| Various | 2 | 0.59% |
| South Africa | 2 | 0.59% |
| Saudi Arabia | 2 | 0.59% |
| Turkey | 2 | 0.59% |
| North Africa | 1 | 0.30% |
| France | 1 | 0.30% |
| Estonia | 1 | 0.30% |
| Poland | 1 | 0.30% |
| Romania | 1 | 0.30% |
| Faroe Islands | 1 | 0.30% |
| Norway | 1 | 0.30% |
| Japan | 1 | 0.30% |
| Croatia | 1 | 0.30% |
| Wales | 1 | 0.30% |
| Sri Lanka | 1 | 0.30% |
| Indonesia | 1 | 0.30% |
| Mexico | 1 | 0.30% |
| Iran | 1 | 0.30% |
| Morocco | 1 | 0.30% |

Most studies aimed to present steps, frameworks and guidelines to integrate the two approaches. Another presented one or more specific hybrid tools that are integrated tools, like E-VSM, that take the VSM from Lean and the environmental aspects from Green. The other major focus of the LG literature is to try to discuss the main links between these approaches, the synergies and trade-offs between them, highlighting the negative and positive impacts that each one has on the other. However, there is no study that discusses LG from the OS perspective.

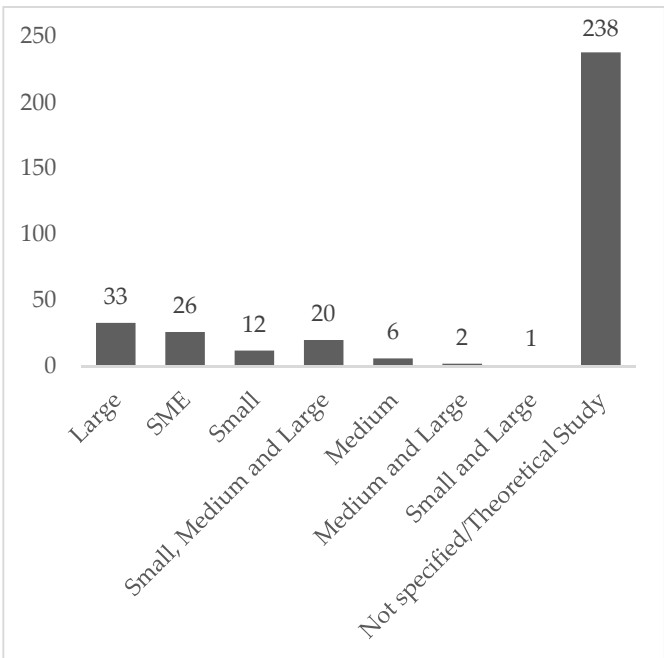

**Figure 9.** Size of the companies studied.

*4.2. Discussion*

First, it is relevant to present the CP and DA that are discussed in the LG literature. The priority "cost" was the most frequent in these studies, and it is related to one of the main goals of Lean Manufacturing, i.e., cost reduction [15,92]. Cost reduction used to be the motivation of companies that aim to become green: they seek to reduce the costs of materials and energy inputs as well as the costs of waste disposal [93].

Regarding the priority "environment", most of the studies discuss the reduction of the negative environmental impacts from the production process that implements Lean. These studies consider this competitive priority as a new one aiming to reduce mainly energy consumption, emissions, water consumption, waste generation and toxic substances. Also, some researchers call the negative environmental impacts in Lean operations "Green waste" or "Environmental waste" [10,11,76,94].

Next, "quality" and "delivery" are both focuses of Lean implementation [46,53]. Quality is discussed, as in [12,95], as a reduction or elimination of defects which can decrease the consumption of raw materials and the costs of production. The priority delivery is addressed when the studies discuss order lead-time, which is one of the metrics considered in Lean implementations. According to Dües et al. [96], customer satisfaction is driven by the reduction of lead time and, as indicated by [30], one of the results from Lean is to allow faster deliveries.

The other priorities found in the studies, but less frequently in Lean systems, were social, service and flexibility. The priority social is related to improving workers' health, safety and morale, improving the local supply, and reducing corruption risk [33]. Service is mentioned as the improvements in customer satisfaction through Lean and Green implementation [96]. Regarding flexibility, the capacity to increase the mix of products that can be targeted by LP is mentioned [31,97].

In addition to the CP, we observed the DA of OS cited in these studies. It is possible to note that Performance Measurement System is the most frequent decision area in these studies. This can be explained because the inclusion of an environmental performance indicator is one of the highest requirements to consolidate GP in industrial operations [18,37,76], and the performance measurement systems in LM are an important aspect [98]. Aligned with this, as highlighted by Leong et al. [6], it is necessary to use support technologies that allow the operational data to be registered for an effective improvement process.

Regarding Supply Chain, the literature discusses mainly supplier relationship (collaboration, selection, purchasing) and logistics [99]; trade-offs or synergies from the JIT deliveries and the integration of the supply chain to implement an LCA; or it analyzes environmental impact from the Life Cycle perspective. Human Resources is another much-discussed decision area, which includes essential changes and the importance of employee involvement in training to implement and to sustain both Lean and Green [30,47,98].

Quality Management is a decision area that is also studied. It includes the implementation of quality programs, like six sigma or Total Quality Control or ISO standard [100]. Technology is normally related to the improvement of equipment to reduce resources consumption, which includes information systems and Industry 4.0 initiatives [6,47]. Product development, on the other hand, addresses improving the design of the product, aiming to improve environmental performance since the beginning of the project [56,96]. Organization area is related to issues of the leadership structure required for Lean and Green integration to attain success in the implementation of the practices [30,33]. Production, Planning and Control discussions include the impacts of a schedule to achieve environmental optimization. Facilities are mentioned to address layout change or facilities projects, like the positive environmental impacts of i cellular manufacturing implementation because of less motion and transportation [101]. Finally, capacity, the least addressed decision area, was discussed regarding the size of batch and capacity planning [9].

Regarding how LG affects the operations' competitive priorities, causing trade-offs synergies and decision area(s) modified with the implementation of LG, we found the relationships between LG and OS are studied in three different ways by the studied authors. The first group focuses on discussing the positive and negative environmental effects of LP on GM; thus Lean and Green are analyzed separately. The other one discusses the GP adopted as a complement to Lean practices by Lean-oriented companies to address the negative environmental impacts of their operations. In this group, Lean and Green are analyzed separately. Finally, in contrast to the previous groups, the third one discusses the effects of Lean and Green together through the analysis of the so-called hybrid practices, or Lean-Green practices, on some Operations Strategy aspect.

### 4.2.1. Lean Practices

It was possible to find a wide range of cited Lean practices, covering 26 different practices in total, as presented in Figure 10. They are used in these studies to show how they can help to achieve better environmental performance, such as by reducing energy consumption, the consumption of materials, wastes generation and emissions; and to show the benefits already known from Lean, such as the reduction of lead time and cost, as well as the improvement of quality and productivity. Moreover, there are some practices, like VSM and TQM, that are mentioned as a foundation for integrating Lean with Green and make a hybrid practice showing synergies between them. These practices are presented in Section 4.2.3. However, some studies mention that there are practices that can increase negative environmental impacts.

As regards the synergy with the priority "environment", all the LPs are considered synergic because of their main focus on waste reduction. Several studies, e.g., [11,102], have demonstrated that Lean Practices can bring environmental benefits and that this can be attributed to the more efficient use of resources (e.g., water and other inputs). Similarly [10,12,41,42,96,103–106] argue that LP implementation can offer significant advantages and synergies with the green performance of companies, which are mainly related to the reduction in consumption of materials, energy and water.

However, some studies show that LP can present certain trade-offs with the priority "environment" [15,52,73,76]. One of the most frequent trade-offs cited refers to delivery and environment. LP may negatively impact the environment, since the JIT (JIT) delivery process results in more deliveries, and then more emissions from the vehicles [5,15,17,47].

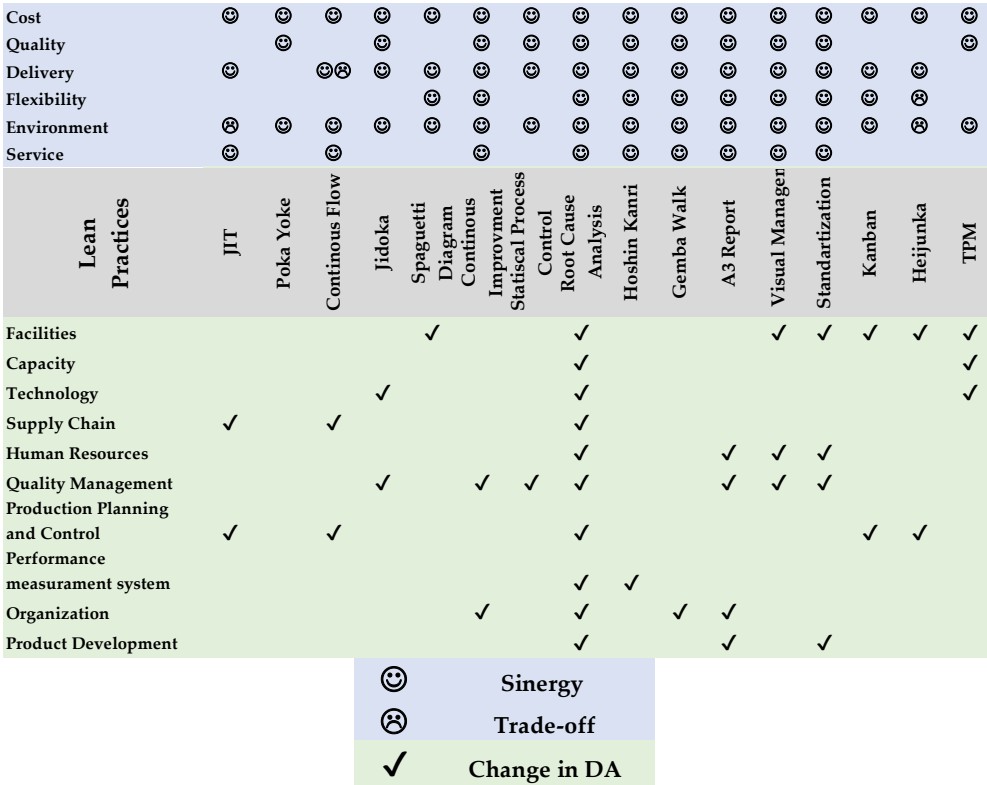

**Figure 10.** Lean Practices in Operations Strategy. Source: created by the authors.

Another trade-off identified is related to environment and quality. Pil and Rothenberg [107] exemplify that sometimes more resources consumption is necessary to maintain a good quality of product. They show that in the paint process, a large amount of water is necessary to have good quality. Therefore, quality-oriented practices such as TQM and Six Sigma can generate this trade-off, which means the reduction of resource consumption can be limited due to technical requirements of the process and the product.

Moreover, flexibility and environment can present trade-offs. Small batches allow more product variety, but they may increase the number of setups [96]. In this sense, to maintain the programming level in variety and volume of items produced, Heijunka is used. Consequently, more setups are necessary. Therefore, as pointed out by [96,103], more cleaning products are required and an increased disposal of unused process material. That way, as exposed by [108], the increase in the disposal of chemical products results in unrecyclable sewage and waste chemical reagents, eventually increasing the environmental burden. In the same way, the increase in consumption of these products can be caused by the practice 5s, which focuses on improving the cleaning and organization of the work environment and on the elimination of unnecessary items which can contribute to quality and cost [23,109,110].

### 4.2.2. Green Practices

Figure 11 presents six practices focusing on reducing the environmental impact in production process, product design and supply chain. They are presented as a complement to LP in companies, or they are part of a framework seeking to integrate Lean and GP. There are also some studies that highlight that it is essential to implement GP in Lean systems to resolve trade-offs with other competitive priorities. Also, there are cases that show their application is done separately from Lean, aiming only to reduce environmental impacts and comply with the legislation.

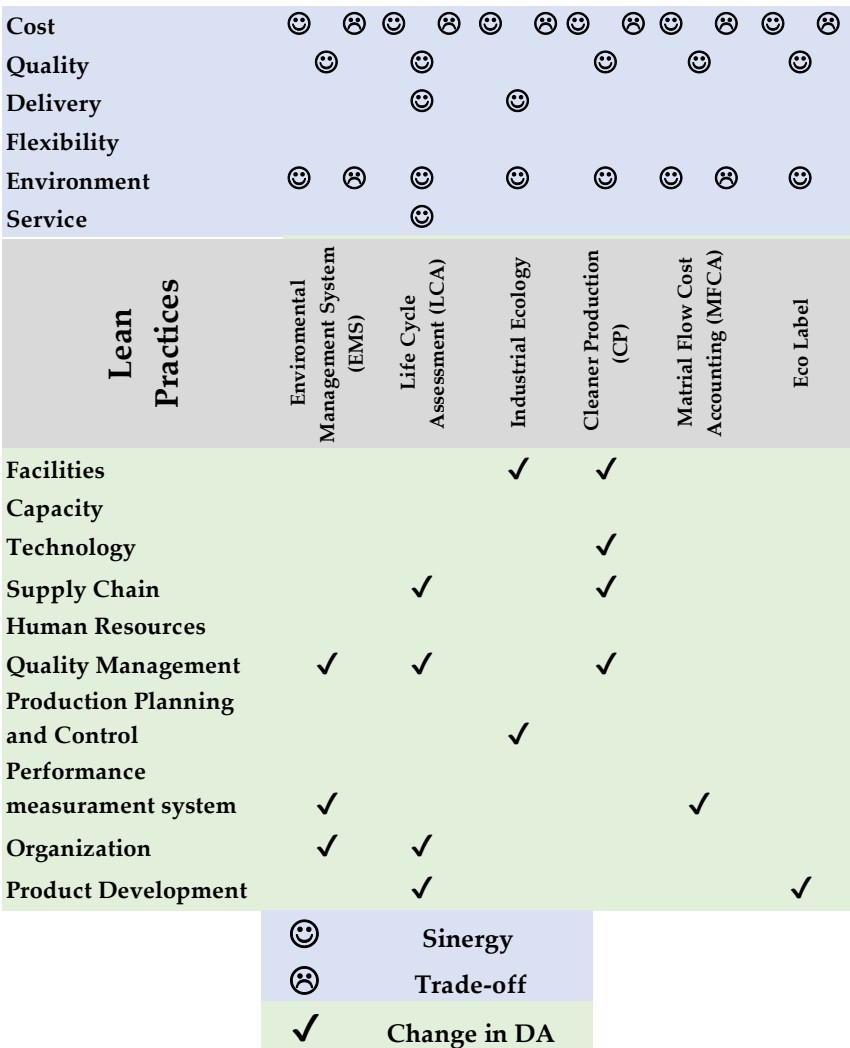

**Figure 11.** Green Practices and Operations Strategy. Source: created by the authors.

In terms of synergies between the competitive priorities from GP, it was possible to note that all of them contribute to and focus on the efficient use of resources; as a result, the priority cost can be improved. It is increasingly related to the cost reduction of material; water and energy consumption as well as waste disposal costs have dramatically increased over the past decade [93]. Furthermore, there are empirical studies that present the cost reduction through improvements in environmental aspects [33,34,111–113].

A study [114] argues that GP can reduce material and production costs, reduce transportation and logistics costs, increase product quality and reduce warehouse costs. Moreover, improving the environment can lead to cost reductions from any punitive or restrictive measures that may be introduced through legislation [46,113]. Mor et al. [48] points out that these practices are not only good for sustainability but also good for business value. Jabbour [23] concludes that the implementation of the EMS, the more frequent practice in the studies, can have positive impacts in various areas of organization performance.

Lastly, it is important to present the practice Material Flow Accounting (MFCA), a method to measure environmental performance by an accounting approach for estimating the output-input ratio and material flow using physical and monetary units [108]. Additionally, Dues et al. [96] argue that GP has a positive influence on LP.

Although several studies argue that there are cost savings with the implementation of GP, it is possible to have certain trade-offs. These conflicts happen when the green improvement means an increase in costs, or requires a huge investment. For example,

when the results from an LCA, or an improvement from an EMS, identify that it is necessary to buy less harmful or more efficient equipment, the financial return on investment can be negative. Another example is related to the supply chain; when to reduce the $CO_2$ emission, the load is consolidated, to seek to transport with the least number of deliveries possible, resulting in fewer emissions and fuel consumption. Thus, it can generate an increase in delivery lead time because some loads would have to wait to fill up the load.

### 4.2.3. Lean-Green Practices

This topic explains the LG Practices and it shows the links with OS in Figure 8. They are described in more detail because they represent a consolidation of LG integration. These practices take the LP as a foundation and include the environment as a target of the improvements. This integration is one of the main steps for a LG practice and it helps to solve the trade-offs. In Figure 12 we can represent the Lean-Green practices relations with trade-offs, synergies and changes in DA.

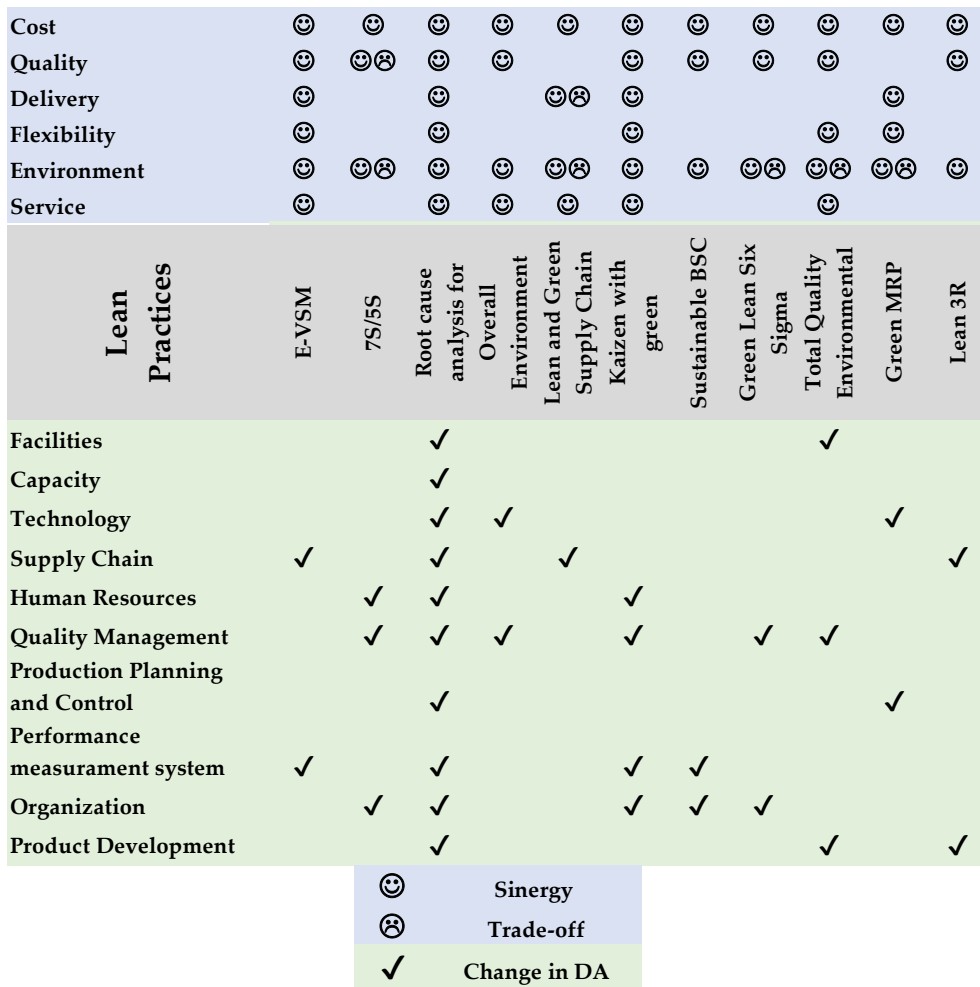

**Figure 12.** Lean-Green Practices and Operations Strategy. Source: created by the authors.

E-VSM (Environmental Value Stream Map)

E-VSM practice, also called Green Value Stream Map, Extended Value Stream Map or Sustainable Value Stream Map, was the most frequent LG practice in the studies analyzed. It aims to map the processes, present the main process indicators, and highlight the problems in the current state. This tool is based on Lean, specifically in VSM, and it was created to guide improvement processes and facilitate communication between those involved. The environmental aspects, such as energy, water and materials con-

sumption and emissions, are included in E-VSM, which helps to collect and analyze these environmental data [76,78,115,116].

Regarding CP, this tool seeks to highlight performance indicators related to delivery, measuring lead-time of operations, quality, showing the rate of rework and defects in operations; and environmental performance indicators such as consumption of materials, energy, water, emissions, among others. Cost is widely cited, since the general objective of an E-VSM is to reduce all types of waste, including environmental waste, seeking cost reduction or, in some cases, to improve operational efficiency [25,37,116,117]. There is no trade-off observed with this practice; E-VSM has demonstrated a very synergic relationship by focusing on both Lean and Green.

In relation to the decision areas, this practice had a great impact on the decision area "performance measurement system". It is fundamental to measure and map the environmental impacts of the processes. The proposal is that sustainability indicators–raw material consumption, water consumption, energy consumption, waste generation, gas emissions, worker safety, ergonomic aspects and noise levels–should be integrated in the traditional VSM, as exposed by Helleno et al. [118] based on the proposal of [56,117,119,120]. In summary, as the study [119] discusses, the E-VSM seeks to provide a wealth of information that can facilitate communication and management through Lean and Green indicators. The study [121] highlights that it is necessary to determine a sustainability indicator set for the E-VSM tool so it would support thee decision making process.

The Supply Chain is another area of decision which is related to the E-VSM practice [122] since it can be used to obtain a global view of the entire chain [56] (Caldera et al., 2017)–from the extraction of the raw material to the final disposal–helping in the measurement, monitoring and visualization of possible improvements throughout the value chain. A point highlighted by [25] is the importance of integrating process information and understanding the client's value, i.e., capturing stakeholders' expectations following a life cycle perspective [3]. Therefore, when considering and integrating the entire value chain, it is essential to use an LCA that contemplates the product/service life cycle 'from cradle to grave', which is the concept that is included in the E-VSM to measure the environmental performance of the entire supply chain [123]. It is also important that circular economy strategies (recycling, remanufacturing, etc.) are considered a driver for the sustainable development. To adopt CE, it would be necessary to understand the productive process and to get information from it [1,124]. Thus, -E-VSM can support CE by drawing the process and presenting the data from it. The data collection and information sharing can be a great barrier to implement this strategy, but the literature has shown that I4.0 strategies can overcome this [125,126].

7s

This practice is an extension of the 5s tool that includes safety in some cases, is called 6s, and can be further expanded with one more s of sustainability. As far as priorities are concerned, this practice is used for cost reduction, quality and environmental improvement through waste reduction and environmental organization [79,110]. Despite the presence of synergies to improve the cost, quality and environment, 7s may create trade-offs when increasing the consumption of cleaning products. Regarding the decision areas, it is noted that Quality Management in the improvement programs and Human Resources in the training are changed with the inclusion of safety and environmental aspects.

In summary, the study presented by [79] shows that this program is the basis for improvement programs in the working environment. The study by [110] states that the objective of this practice is to improve quality, increase sales, reduce cost and provide a quality environment. Duarte and Cruz Machado [19] explain that this is a standardized work methodology used by organizations to collaborate to achieve Green objectives.

Root Cause Analysis for Environmental Problems

According to [109], the practice of root cause analysis can be extended to identify potential causes of environmental problems in processes, as already done for problems related to other competitive priorities.

Pinto and Mendes [127] point out that this practice should be structured to allow the visualization of environmental problems and should be structured in cycles and improvement programs, such as PDCA and Kaizen. Galeazzo et al [45] comment that by adopting this approach together with quality managers it was possible to understand and question environmental problems and thus identify possible solutions. In summary, these studies infer that just like recurrent problems of quality or process inefficiency to reduce cost, environmental problems should be solved using the same reasoning as the others, and they can be included as variables for the cause of the other problems. Thus, it was found that these practices can change the way to solve the problems in Quality Management and support the achievement of cost, quality and environment.

OEEE

OEE is an abbreviation for the term Overall Equipment Effectiveness, which means the overall efficiency of the equipment in dealing with an indicator that monitors the efficiency of the manufacturing process. The OEEE, according to Domingo and Aguado, 2015, means Overall Environmental Equipment Effectiveness, and it incorporates the concept of sustainability based on a calculation of the environmental impact in the life cycle. According to the authors, using this indicator can allow decision making by integrating sustainability and making a comparative analysis of environmental impact when analyzing improvements implemented in the processes of the organization. Thus, it was observed that OEEE has a relationship with the area of Performance Measurement System, and aims at efficiency improvements in which one monitors the performance in productivity and quality that imply cost reduction.

Further, this hybrid indicator is linked and can change the LP TPM and SMED. The first practice, TPM (Total Productive Maintenance), is a set of techniques to ensure the reliability and productivity of all machines in the production process [128]. The SMED (Single-Minute Exchange of Dies) or Quick Tool Change has as a principle to perform setup in less time, allowing the increased flexibility and/or availability of the machine in the production flow.

Lean and Green Supply Chain

This practice, presented in the study by Sant'Anna et al. [129]), was defined as the combination of the three approaches Lean, Green and Supply Chain, seeking cooperation for cost reduction, consumer focus, quality and environmental management through the ISO9000 and ISO14001 standards and risk management. In addition, the authors emphasize that the integration of the three approaches must be done to meet legal requirements, since Lean may not be sufficient to achieve them, and environmental management can collaborate in this direction. Thus, this practice seeks to include environmental performance in the decision area of supply chain and quality, as well as the objective of cost, quality and environmental performance [97]. This practice also involves supplier selection, procurement, third-party logistics and transportation, that aim to minimize the environmental impact of the product [17].

One practice that is widely used to improve the delivery and reduce cost in the supply chain is the JIT. As we cited before, this practice can present trade-offs between the competitive priorities "delivery" and "environment" because the studies show that to be better in delivery it is necessary to increase the consumption of fuels and then increase emissions for the environment [15,17,49,52]. Furthermore, it is important to highlight that LCA is a widely-used practice to measure environmental impacts in the supply chain. Thus, the decision area "Supply Chain" is modified by the practices Green Supply Chain, JIT and

LCA, and has to start to consider the trade-offs between delivery and the environment in its decisions.

Kaizen with Green Goals

Lean uses kaizen for improvement processes. In this search, studies have been identified that deal with kaizen integrated with GP. They show that kaizen can be integrated into ISO14001 continuous improvement cycles, which can help involve employees and find innovative solutions to the problems identified [56,130].

Kaizen with Green goals can affect mainly, in a synergic way, cost, quality and environment, because it includes environmental objectives that can reduce energy and material consumption. Thus, environmental metrics, such as energy consumption (analyzing monthly energy bills), and water consumption within a period, are used to determine costs [76]. In summary, as US EPA [131] shows, Kaizen activities should be aimed at reducing environmental costs in addition to tracking them, and that the tools used in the improvement cycle are mainly traditional Lean tools. Thus, this practice implies changes in improvement programs (Quality Management) and training for human resources (Human Resources), and more involvement of employees and the inclusion of the environment as a priority in the implemented improvements, measured through environmental performance indicators.

The study about a Lean and Green model [76] highlights that Kaizen is a great way to promote green improvements in organizations. As a study of Oliveira et al. [132] shows, Kaizen is a practice characterized by the involvement of staff for a specific goal, and can be a good strategy to identify and improve environmental aspects.

Lean with Environmental Indicators

First, it is important to show the study by [16] that shows an adaptation in the BSC strategic planning tool which includes sustainability in financial, client, internal processes and learning and growth perspectives: SBSC—Sustainable Balanced Score Card. In summary, the BSC is about several interrelated indicators, and indicators from these perspectives. According to the authors, including all these aspects in strategic planning can enable better performance management of the organization by senior management, and can be a basic model to structure the integration of Lean and Green. The study of these authors highlights the inclusion of the social perspective, which is not the focus of this research, and the environmental perspective.

The authors stress the importance of performance measures and incentives for investments in information systems, coordination and autonomy for decision making. Thus, the use of this model for LG integration changes the system of performance measurement, information systems (technology) for measuring, recording and sharing information, and it seeks to monitor the results obtained in all competitive priorities and in the value chain from "Cradle to Grave".

In addition, the most frequent indicators related to environmental priority are energy consumption (kw/h), water consumption ($m^3$ consumed per period), kg of materials consumed, kg of waste, kg of $CO_2$ emitted, among others. Several studies highlight the importance of including environmental performance indicators, since this is what will allow the integration with lean systems and the monitoring of environmental gains [17,33,37,76,118,133].

As we mentioned in the E-VSM topic, the I4.0 tools have supported the performance measurement. An example is the technology of wireless measurement systems used to capture data in industries, which collect accurate information such as energy consumption and costs, water consumption, gas emission, etc. [7,29]. From the analysis of how much is being consumed it is possible to obtain a more precise knowledge of the processes in relation to critical resources and thus make improvements attacking the root cause of the problem [7].

### Lean 3R

Lean 3R is related to the remanufacturing, recycling and reuse of products and/or materials used in the processes [134]. It is defined as a product recovery process that uses energy and available resources, reduces the waste associated with the processes, and therefore can increase the overall efficiency of the process. The advantages associated with it can include the reduction of lean waste, such as overproduction, inventory, lead-time, unnecessary movement, waiting time and transportation [135]. Thus, there is a synergy between the environment and cost. To be able to apply lean remanufacturing, this must be considered from product and process Design [136]. The GP Design for Environment and the LCA are very useful for this practice [15]. Moreover, this is very much aligned with the strategies of the Circular Economy or Circular Manufacturing, which can help to reduce resources consumption and to extend lifecycles by remanufacture, recycle, resource efficiency, waste management, etc. [1].

### Green Lean Six Sigma

This practice is a methodology that allows the search for environmental performance, applying robust tools for analysis and problem solving. Green Lean Six Sigma utilizes traditional aspects of Lean and Six Sigma while providing the tools needed to identify, implement and structure improvements that have a positive impact on the environment [115].

It is based on six pillars: leadership and people, Green and Lean six sigma tools, continuous improvement, strategic planning, interaction with stakeholders and results, and knowledge management; refs. [33,137] highlight that this proposal has a great impact on product development, contributing to cost reduction, process optimization, and enabling sustainability. Ruben et al. [84] demonstrate that the Lean Six Sigma basis allows the reduction of process variations and thus helps in the reduction of defects and waste generation during the production process. In addition, the study by Sreedharan V et al. [133] indicates that the use of these concepts contributes to the competitive priorities cost, quality and environment, through the reduction of environmental impacts and increases in the level of service provided. It is also noted that this practice mainly modifies the following decision areas: Quality Management and Human Resources.

### Total Quality Environmental Management—TQEM

This practice is a sub-development of the practice Total Quality Management- TQM, which extends the principles of quality management to include manufacturing practices and processes that affect environmental quality [12,15,34,47,107]. TQM can help increase quality [35] and reduce production defects, which consequently reduce the consumption of raw materials and energy use [95,138]. However, trade-offs may arise when the production systems need more resources to improve the quality of the product [20,107].

### Green MRP

The Green MRP tool is essentially a conventional Material Requirements Planning system that has been modified to include environmental considerations with the objective of minimizing the environmental impact of the generated waste, and seeking to increase the planning potential for the components and process waste, i.e., optimizing production planning in order to reduce possible problems related to the environment [139].

This practice is directly related to the decision area "Production Planning and Control". The production schedule and delivery schedule can help environmental performance by minimizing net energy consumption and defining shorter delivery routes to reduce the emissions of $CO_2$ [46]. Thus, this tool aims to balance better production planning with cost and environmental performance. Therefore, this tool directly influences the decision areas "Production Planning and Control", "Capacity" and "Technology".

Closing Remarks from Lean-Green Practices

The main aspects from the hybrid practices that integrate Lean and Green are that they focus on simultaneously considering the environmental performance and the other performance goals with the same degree of weighting in most cases, which helps to solve the trade-offs that can emerge in production systems. Because these tools make environmental indicators visible, and train employees in environmental priorities, they make the environment an important variable in the process of continuous improvement and decision making.

Another important aspect from hybrid tools and LG integration is that simultaneous implementation of LG methods multiply performance parameters, and this can result in significant cost reduction [111]. Furthermore, when Lean and Green are implemented together, they can create a more significant positive impact on organization than when implemented separately [96,140].

Finally, it is important to observe that the hybrid LG tools emerge from Lean. This feature can infer that Lean is used as a foundation for the management of production systems to implement the OS. Furthermore, GP works as a complement of Lean to help to support the OS that the environment is a competitive priority. Lean can also be insufficient to be fully Green.

*4.3. Summary and Research Agenda*

Most of the studies found are empirical and focus on understanding the relationship between Lean and Green and how they can be integrated. Their results suggest that Lean and Green objectives are different in essence, i.e., while LP aims to reduce cost and lead time (delivery) and to improve quality and flexibility, GP seeks to reduce waste related to environmental impacts [20,33,48,49,53,111,117]. Fahat et al. [141] cite that Lean waste is different from Green waste. Pinto and Mendes [127], in agreement with [73], point to the fact that Lean and Green objectives are different and generate different impacts on business performance. GP has the direct and clear objective of reducing the environmental impact of processes, while LP will directly impact on cost reduction, lead time and quality improvement; and the improvement of environmental aspects can be achieved indirectly.

From these results it is possible to understand that Lean and Green are synergic in most of their practices, but some trade-offs exist. In this sense, it is important to have the OS well defined to support the strategy and the targets of the organization. When the environment is a CP, Lean can be insufficient to solve the trade-offs; thus it becomes necessary include GP directly in the decision areas. Further, when LP and GP are implemented together, it becomes possible to leverage the performance of an organization more than when they are implemented separately.

Moreover, these results reveal to us that there are many trade-offs that have not yet been explored, as well as synergy relationships that are often limited to waste reduction and efficient use of resources. Furthermore, the strategic perspective of LG is still missing. Given these gaps, future research is suggested:

- Verifying the existence of unidentified trade-offs between the Lean and Green, as well as understanding how, and how much, the competitive priorities are affected; and evaluating quantitatively the relationship between individual practices and the contribution for the competitive priorities.
- Exploring in-depth the changes and the contribution from LG implementation in the decision areas; and exploring the particularities of each practice to be implemented.
- Empirical studies examining how companies frame their competitive priorities and decision areas in different levels of Lean and Green implementation and in different industrial sectors. Further, the researchers in operations management may quantify the performance of the companies related with the operation strategy adopted.
- Quantifying the synergy between the LG empirically, exploring different industrial sectors as well as different operations strategies.

- Identifying and comparing the reasons for OS formulation and Lean-GP adoption, and discussing their strategic role.
- Understanding how the Circular Economy strategies can be supported by Lean and Green Manufacturing practices.
- In terms of research methods, based on theoretical propositions identified qualitatively by most of the researched literature, quantitative approaches can be developed, such as surveys and modeling, to confirm them.

**5. Conclusions**

The main intention of this work was to provide, from studies reported in the literature about LP and GP, an initial and holistic analysis of the adoption of such practices from the perspective of the OS, presenting synergies and trade-offs between competitive priorities; as well to discuss the main changes in decision areas from LG implementation.

The study of these links between the entire content of the OS in the LG literature is still recent and complex. The systematic review of the literature performed indicates that the analyzed articles do not cover the entire content of the OS and that the studies are still exploratory. As discussed, there is a tendency to favor a few practices, or a few competitive priorities; or only economic results and environmental impacts, namely in terms of cost reduction and energy consumption, waste generation and emissions resulting from the implementation of LG. Through this systematic literature review, it was possible to answer the proposed RQs based on a content analysis performed with 338 papers. The results of this article have the potential to help managers and policymakers gain a holistic understanding of how they can implement integrated practices and learn about existing practices to improve the environmental impact of lean systems.

This work has some limitations related to its search strategy, including the three selected databases and only journal publications in English. In addition, it focuses only on the immediate links between the content of OS and LG, excluding the enabling and hindering variables of practices and OS adoption.

**Author Contributions:** Conceptualization, G.A.Q., I.D. and A.G.A.F.; methodology, L.A.d.S.-E. and A.L.V.T.; validation, G.A.Q. and I.D. analysis, G.A.Q. and A.G.A.F.; investigation, G.A.Q. and A.G.A.F.; resources, G.A.Q., A.G.A.F., I.D. and L.A.d.S.-E.; writing—original draft preparation, G.A.Q., A.G.A.F., I.D. and L.A.d.S.-E. and A.L.V.T.; writing—review and editing. All authors have read and agreed to the published version of the manuscript.

**Funding:** This research was supported by Coordination for the Improvement of Higher Education Personnel (Coordenação de Aperfeiçoamento de Pessoal de Nível Superior CAPES), scholarship number 88887.194801/2018-00, 88887.514124/2020-00 and 88882.426297/2019-01.

**Institutional Review Board Statement:** Not applicable.

**Informed Consent Statement:** Not applicable.

**Data Availability Statement:** Not applicable.

**Conflicts of Interest:** The authors declare no conflict of interest.

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
