# Peer review of "Synergies and Trade-Offs between Lean-Green Practices from the Perspective of Operations Strategy: A Systematic Literature Review"

_sustainability, doi:10.3390/su15065296_

Round 1
Reviewer 1 Report
I appreciate the effort involved for this study and congratulate the authors for this, but from my point of view the manuscript is in danger by self-plagiarism (more than 80%). In order not to deceive (first the editors, reviewers and then readers) properly quoted of the PhD thesis “ESTRATÉGIAS DE OPERAÇÕES E PRÁTICAS LEAN-GREEN: UM ESTUDO DE CASOS MÚLTIPLOS EM EMPRESAS DO SETOR AUTOMOTIVO”, cited and correct acknowledgment shell be use in the manuscript. Some materials can be reused but, if there are more coauthors, I suggest updating the material as well as the references that were left in 2020. Success!
Author Response
Dear reviewer,
It is attached the reply.

Reviewer 2 Report
From my point of view, this is interesting review, which tries to describe the connection between operational strategies and lean-green approaches. At the same time, it requires some modification to be suitable for publication.
1.First of all, i don't understand, why authors limited their research by Nov 2020. Since their analysis showed that the half of the articles was published in 2019 and part of 2020, it is possible that similar number of studies was published in 2021 and 2022. It is too huge loss for 2023 literature review.
2.Section 3.4. i recommend to add details about each step of analysis.
3.Figures. It is not necessary to duplicate title in figure and in caption, but it is necessary to add labels for axis.
4.Fig. 9-11. I strongly recommend to interpret these data in another manner. From these figures i see only that almost everything is connected with almost everything. And it is also necessary to add numerical data here. For example, you can use heatmaps for that.
5.Section 4.2.3. It is necessary to extend the description of practicies with adding analysis of their strengths and weaknesses.
Author Response

(The authors gave the same response as above.)

Reviewer 3 Report
The paper presents an exceptionally comprehensive study on the evolution of the Lean-Green agenda in the academic literature throughout recent decades. The relevance of the topic and the gaps in current research are convincingly demonstrated in the introduction. The findings are presented in a logical manner, as well as they are discussed through the lens of previous studies. The contributions of the study to the literature are evident.
Nevertheless, the current version of the manuscript needs certain revisions in terms of the presentation of the methodological approach and the format of references.
Line 199: Only papers in journals? Were books, monographs, chapters, conference proceedings, reports, etc. included? If not, why?
Lines 226-227: How exactly were the duplicates excluded? Manual check across the array of over 5,000 titles? The authors should detail the approach, as it may be helpful for other scholars in future studies.
The style of In-text citations must be unified. According to the Sustainability guidelines, the following format must apply: Leong et al. [6], not Leong et al. (2020) - see in Line 44. The entire text must be checked.
Author Response

(The authors gave the same response as above.)

Round 2
Reviewer 1 Report
I consider you have addressed the suggestions, and the result is satisfying. Thank you and wish you great success in your endeavours!
Reviewer 2 Report
I would like to thank authors for their work and recommend this manuscript for publication.